# Reliability Evaluation of Virtual Power Plants Based on Bayesian Networks

Jiacheng Hu
School of Electrical Engineering and Automation
AnHui University
HeFei China
2692177751@qq.com

Ning Zhang*
School of Electrical Engineering and Automation
AnHui University
HeFei China
zhangning@ahu.edu.cn

LingXiao Yang
School of Artificial Intelligence
AnHui University
HeFei China
yanglingxiao@ahu.edu.cn

Cungang Hu
School of Electrical Engineering and Automation
AnHui University
HeFei China
hcg@ahu.edu.cn

*Abstract*—**When utilizing traditional reliability analysis and evaluation methods to assess the reliability of virtual power plants, there are numerous shortcomings: the lack of intuitive representation of probabilistic relationships, difficulties in integrating and processing uncertain factors, leading to limited reasoning and fault diagnosis capabilities. To overcome these deficiencies, a reliability evaluation method for virtual power plants based on Bayesian network approaches is proposed. This method not only accurately calculates the reliability indices of virtual power plants but also visually demonstrates the extent of influence that individual components or combinations of components have on overall reliability. Firstly, a Bayesian network is employed to construct the reliability model of the virtual power plant. Secondly, the Bucket Elimination method is utilized for precise probability calculations. Finally, in conjunction with the Monte Carlo method, a comprehensive analysis of the reliability of the virtual power plant is conducted.**

*Keywords—Fault Analysis of Virtual Power Plants; Bayesian Networks; Reliability; Monte Carlo Method*

## I. INTRODUCTION

In the current trend of energy transformation, virtual power plants (VPPs) have emerged as an innovative technical solution aimed at integrating and optimizing various types of distributed energy resources to enhance the flexibility and sustainability of power systems[1]. The birth of this concept stems from the interplay of multiple socio-economic and technological factors. On one hand, the rapid growth of renewable energy sources, particularly wind and solar power, poses stability challenges to traditional power systems, necessitating more sophisticated supply-demand management and dynamic adjustment capabilities. On the other hand, the widespread deployment of distributed energy resources, such as small-scale wind turbines, rooftop photovoltaic systems, and energy storage devices, has led to the emergence of the "prosumer" role, where users are both electricity consumers and potential energy producers. Simultaneously, the development of smart grids and advancements in information technology provide the necessary technological support for the implementation of VPPs, enabling real-time monitoring and dispatch of decentralized energy sources through advanced algorithms and communication networks. Furthermore, the demands for environmental protection and changes in energy markets have underscored the role of VPPs in improving energy efficiency, reducing carbon emissions, and promoting the development of a green economy[2-4].

As a core component of the energy transition, the role of virtual power plants cannot be underestimated, yet they also face a series of substantial challenges. The primary challenge lies in managing the inherent intermittency and uncertainty of renewable energy sources, such as wind and solar power, which are greatly influenced by weather and time. This requires VPPs to possess rapid response capabilities to instantly adjust the balance between power supply and demand, ensuring the stable operation of the grid. Another challenge stems from the widespread distribution and diversity of distributed resources, including small wind turbines, rooftop solar panels, household battery storage systems, and electric vehicles, which are scattered across various locations, increasing the complexity of scheduling and integration. VPPs must leverage advanced information technology and intelligent algorithms to effectively manage and coordinate these resources, thereby enhancing the overall system's efficiency and reliability. Therefore, ensuring the reliability of VPP operation is paramount [5].

Scholars from both domestic and international circles have conducted research on the reliability assessment of VPPs. Reference [6] effectively describes VPPs through a cluster-level modeling approach, addressing issues related to the diversity of distributed resource entities, significant characteristic differences, and the diversification of grid demands, presenting new solutions. Reference [7] delves into how power information systems impact primary power systems, particularly their direct and profound influence on power system reliability, offering a fresh perspective for stable power system operation. In [8], researchers propose a new time-varying outage probability model for components, which comprehensively considers factors such as equipment aging and imperfect preventive maintenance. Through different maintenance strategies, new assessment results for system reliability are presented. Reference [9] systematically sorts out reliability indicators for both the physical and information

systems of distribution networks, while exploring the construction of reliability indicators under mutual coupling, providing new insights into distribution network reliability analysis. For smart substation protection systems, [10] constructs a novel reliability assessment model that considers both equipment aging and incomplete planned maintenance, offering a more comprehensive perspective on reliability assessment. From a network structure perspective, [11] proposes an innovative algorithm for grid optimization in VPPs, which is expected to play a significant role in grid optimization. Based on the characteristics of cyber-physical power systems, [12] innovatively abstracts them as unweighted undirected heterogeneous graphs and establishes a new "one-to-one" coupling model, providing a new tool for system analysis. When exploring coupling methods, [13] delves into various connection modes, including one-to-one, one-to-many, and many-to-many, offering new perspectives on understanding system complexity. By introducing the concepts of weighted betweenness and weighted degree, [14] further reveals the core role of high-betweenness and high-degree information nodes in system structural and functional integrity. Considering the importance of both information flow and energy flow nodes, [15] provides a new comprehensive evaluation method for identifying critical nodes in information-energy systems, contributing to enhancing system stability and efficiency.

As a probabilistic graphical model, Bayesian networks excel at handling uncertainty by visually representing conditional dependencies between variables through directed acyclic graphs, making them particularly suitable for analyzing complex systems. This characteristic is particularly prominent when applied to the reliability assessment of VPPs. Specifically, Bayesian networks can capture and quantify various random factors that affect VPP reliability, such as weather changes and equipment failure rates, converting them into probabilistic models for more accurate system state predictions. Furthermore, they are adept at revealing causal relationships within the system, helping to identify key resources or links that have the most significant impact on overall reliability, providing decision support for optimizing scheduling strategies and preventive maintenance. Importantly, Bayesian networks possess dynamic updating capabilities, enabling them to adjust probability estimates based on real-time data, ensuring that assessment results keep pace with actual operating conditions, injecting powerful momentum into the continuous optimization and risk management of VPPs.

This paper first analyzes the structure of VPPs and the key factors affecting their reliability. Secondly, it constructs an analysis model based on Bayesian networks to quantify the reliability of VPPs. Then, using Monte Carlo simulation methods, a detailed assessment of VPP reliability is conducted. Finally, combining the Bayesian network model with instance simulations, a comprehensive analysis of the reliability characteristics of VPPs is presented.

## II. THEORY

### A. Virtual power plant

As the global energy mix undergoes profound adjustments and renewable energy technologies advance rapidly, the large-scale integration of distributed energy sources such as photovoltaic power generation and wind power into the grid poses unprecedented challenges to the stability and supply-demand balance of power systems. Against this backdrop, the concept of the virtual power plant (VPP) has emerged. It aims to integrate various distributed energy resources, including distributed generation units, controllable loads, and energy storage systems, through an advanced distributed management system. These discrete energy components are aggregated into a logically unified and flexibly controllable energy collective. This collective can participate as an independent entity in the operation and dispatch of the power grid, thereby effectively coordinating the complex relationship between smart grids and distributed energy sources and optimizing the overall performance of the power system.

### 1) Structure

Figure 1 depicts the operational framework of the virtual power plant. Diversified energy production, encompassing photovoltaic, wind power, and gas-fired generation, provides the system with a stable and diverse energy supply. Subsequently, electric vehicles act as mobile energy storage units, collaborating with microgrids to form a flexible energy reserve and dispatch network, enhancing the system's energy storage capacity and power supply flexibility. The control center serves as the intelligent dispatching hub, leveraging the big data processing capabilities of the cloud center to manage various energy sources with precision. Ultimately, these energy sources enter the market through electricity trading, achieving efficient energy utilization. The smooth interaction of information flow and energy flow ensures real-time balance between supply and demand. The entire process forms a logically rigorous closed-loop system for energy management and trading, where every link from energy production to final trading is intricately connected, collectively constituting the efficient and intelligent operational system of the virtual power plant.

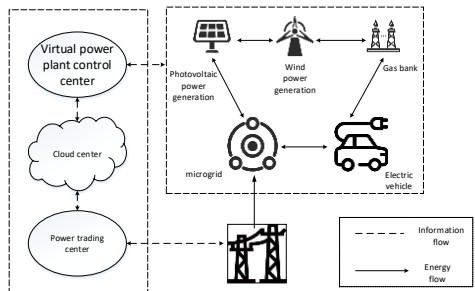

Fig 1 Virtual power plant operation framework

From the perspective of power systems, their normal operation and control are inseparable from the real-time and reliable data transmission support provided by information systems. Conversely, when information systems perform tasks such as data collection, transmission, and processing, they rely on the stable energy supply from power systems. The two are not isolated entities but rather interdependent and develop together. The virtual power plant (VPP) is a product of the deep integration of power grids and communication networks, forming a vast and complex system. However, this complexity also introduces numerous risk factors, as the uncertainty and unpredictability of network failures can potentially impact the

security of the VPP, even leading to power outages. Therefore, conducting reliability assessments for VPPs is of paramount importance, providing robust protective measures to ensure their stable operation.

### 2) Reliability influencing factors

The virtual power plant (VPP) system, as an advanced integrated energy management framework, is essentially a profound integration of physical power systems and informational architectures, aiming to create an efficient and intelligent energy operation platform. Within this system, the reliability of the power system serves as the foundational support, comprising several crucial components: circuit breakers, which act as the core of power protection and are vital to the safe operation of the grid; generators, the starting point of power production, whose stable operation directly impacts the overall system's power supply capability; and transformers, the key points for voltage conversion, ensuring the efficiency and safety of power transmission.

The role of the information system is equally significant, functioning as the nervous system of the VPP, responsible for data collection, analysis, decision-making, and control. Intelligent Electronic Devices (IEDs) play a pivotal role in this process, monitoring and controlling the status of power equipment in real-time to ensure the precision and flexibility of power system operation. Switches, as the central hub of information transmission, guarantee high-speed and accurate data flow, essential for the normal functioning of the information system. Communication lines, the connecting bridges between key nodes, directly influence the timeliness and reliability of information transfer based on their stability and bandwidth.

The overall reliability of the VPP depends on both the performance of physical power facilities and the design and implementation of the information system. Only when these two aspects achieve optimal synergy can the VPP fully unleash its potential, realizing efficient energy management and optimized distribution while ensuring the continuity and quality of power supply, thereby providing more stable, efficient, and environmentally friendly energy services to end-users. This comprehensive management strategy holds profound significance for driving the transformation and upgrading of the energy industry and promoting sustainable development.

As depicted in Figure 2, this paper selects "VPP failure" as the core analytical event to explore its reliability. The power system of the VPP encompasses various switching devices such as circuit breakers, transformers, and generators. Meanwhile, its information system comprises critical components like IEDs, switches, and communication lines. Failures in any component within either the physical or information system have the potential to trigger an overall VPP failure. Based on these logical associations, we have constructed a failure model with "VPP failure" as the top-level event.

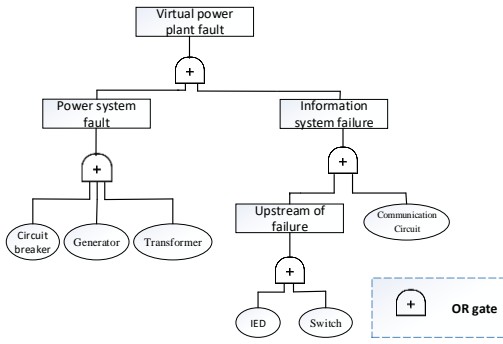

Fig. 2 Virtual power plant fault tree

Based on references [16-17], we obtained the failure rates of individual basic components. Assuming that the reliability of the components follows an exponential distribution, we calculated the estimated monthly failure probability for each device. The detailed data are presented in Table 1.

TABLE I. BASIC COMPONENT FAILURE PROBABILITY

| Parameter | Failure Probability | Parameter | Failure Probability |
|---|---|---|---|
| Circuit Breaker | 0.134 | IED | 0.004 |
| Generator | 0.232 | Switch | 0.001 |
| Transformer | 0.047 | Communication Circuit | 0.006 |

### 3) Reliability index

#### a) Failure rate

A measure of the likelihood that a system or component will fail within a given period of time.

#### b) State Variable

$X_i(t)$ -Describes the state of the i-th component at time t: If the variable's value is 1, it indicates that the component is operating normally at time t; if it is 0, it indicates that the component has failed.

$$X_i(t) = \begin{cases} 1\,(\text{Component normal operation}) \\ 0\,(\text{Component failure operation}) \end{cases} \quad (1)$$

$S(t)$ -Describe the state of the virtual power plant at time t: If the value of this variable is 1, it indicates that the virtual power plant runs normally at time t; If the value is 0, the virtual power plant is faulty.

$$S(t) = \begin{cases} 1\,(\text{Normal operation}) \\ 0\,(\text{Fault operation}) \end{cases} \quad (2)$$

#### c) Mean time to failure (MBTF)

$$MBTF = \frac{T}{n} \quad (3)$$

Where T is the total operating time of the virtual power plant and n is the number of failures.

### B. Bayesian network

#### 1) summarize

Bayesian networks, also known as Bayesian directed acyclic graphs (DAGs) or probability dependence networks, are probabilistic graphical models based on Bayes' theorem. They

utilize directed acyclic graphs to represent conditional dependencies between random variables and quantify these relationships through conditional probability distributions. Constructed on the foundation of a directed acyclic graph (DAG), Bayesian networks consist of a definitive topological structure, nodes, and directed arcs connecting these nodes.

In the simple Bayesian network illustrated in Figure 3, each node represents a variable within the domain, while the directed arcs visually demonstrate the interactions and probabilistic influences between these variables. Through this graphical representation, Bayesian networks effectively communicate uncertain knowledge while clearly annotating conditional probability distributions, thereby accurately reflecting the local conditional dependencies between variables within the model. This representation method endows Bayesian networks with high flexibility and powerful analytical capabilities when dealing with complex systems.

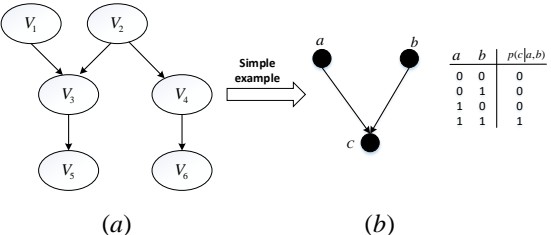

Fig 3. Bayesian network structure

### 2) Inference algorithm

Bayesian networks are capable of not only performing forward reasoning, which involves deriving outcome probabilities from known conditional probabilities, but also backward reasoning, where the probabilities of causes are inferred based on observed outcome probabilities. Among the inference algorithms for Bayesian networks, there are both exact inference algorithms and approximate inference algorithms. Among them, bucket elimination, as a type of exact inference algorithm, is applicable to both singly connected and multiply connected networks. This method meets the analytical requirements of this paper in terms of computational efficiency and reasoning accuracy. Therefore, we choose to adopt bucket elimination to evaluate the reliability of virtual power plants.

### 3) Bucket Elimination

Given a set of random variables X, which contains n variables, $X = \{x_1, x_2, \cdots x_n\}$. For any variable $x_i$ in the set, we can calculate its probability of being in a particular state by the formula (1).

$$
\begin{aligned}
P(X_i = x_i^k) &= \sum_{X|X_i} P(X_1, X_2, \cdots, X_i = x_i^k, \cdots, X_n) \\
&= \sum_{Pa(X_i)} P(X_i = x_i^k, P_a(X_i)) \\
&= \sum_{Pa(X_i)} P(X_i = x_i^k | P_a(Xi)P(P_a(X_i)))
\end{aligned}
\tag{4}
$$

In the formula, $X | X_i$ refers to the set of variables other than $X_i$.

With Bucket Elimination in Fig 3(b), the normal working probability P (c=1) of system node c is calculated as follows:

$$
\begin{aligned}
P(c=1) &= \sum_{a,b} P(a,b,c) = \\
&\sum_a P(a) \sum_b [P(c=1|a,b)P(b)] \\
&= P(a=0)P(c=1|a=0,b=1)P(b=1) \\
&+ P(a=1)P(c=1|a=1,b=1)P(b=1) \\
&= P(a=1)P(b=1)
\end{aligned}
\tag{5}
$$

## III. VIRTUAL POWER PLANT RELIABILITY ASSESSMENT

### A. Bayesian network establishment

Using the method described in reference [18], we converted the fault tree in Figure 3 into the corresponding Bayesian network. The specific mapping result is presented in Fig 4.

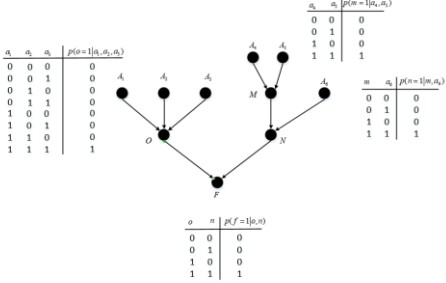

Fig 4. Bayesian network model of virtual power plant

As shown in Table 1, the probability of normal operation of the virtual power plant is calculated by using the bucket elimination method as follows:

$$
\begin{aligned}
P(f=1) &= \sum_{a_1,a_2,a_3,a_4,a_5,a_6,o,n,m} P(a_1,a_2,a_3,a_4,a_5,a_6,o,n,m) \\
&= P(f=1|o,n) \sum_{a_1,a_2,a_3} \left[ p(0|a_1,a_2,a_3) p(a_1) p(a_2) p(a_3) \right] \\
&\cdot \sum_{m,a_6} \left[ p(n|m,a_6) p(m) p(a_6) \right] \\
&\cdot \sum_{a_4,a_5} \left[ p(m|a_4,a_5) p(a_4) p(a_5) \right] \\
&= P(a_1=1)P(a_2=1)P(a_3=1)P(a_4=1)P(a_5=1)P(a_6=1) \\
&\approx 0.63
\end{aligned}
\tag{6}
$$

### B. Reliability calculation process

This paper employs the Monte Carlo simulation method to quantify the operational reliability of virtual power plants. Random sampling is conducted for the potential failure scenarios of the six key components that constitute the virtual power plant, in order to obtain the potential failure interval data for each factor. During the f rounds of simulation experiments, random samples are independently drawn for the failure times of each reliability indicator.

The state variable of parameter i at time t is:

$$X_{if}(t) = \begin{cases} 1(\text{Component normal operation}), t<t_{if} \\ 0(\text{Component failure}), t> t_{if} \end{cases} \quad (7)$$

Then the state variable of the virtual power plant at time t is:

$$S_f(t) = \begin{cases} 1(VPP \text{ is operating normally}), t < t_{if} \\ 0(VPP \text{ fault}), t> t_{if} \end{cases} \quad (8)$$

The pseudo-code of the calculation process is shown in Table 2.

TABLE II.     CALCULATION PROCESS OF RELIABILITY INDEX OF VIRTUAL POWER PLANT

**Algorithm 1: Pseudo-code for Training and Calculating Virtual Power Plant Reliability Indices**

```
Input: Tmax = Total set runtime for the virtual power plant;
    m = Number of equal time intervals divided;
    f = 1, Initialization of simulation count;
    M = Total number of maintenance activities;
Output: Reliability Indices
begin
  while f <= Set number of simulations do
    f = f + 1
    for each reliability parameter do
      failure_time = RandomNumberGenerator()
      Adjust_failure_time_with_new_parameter_X(failure_time)
      Add_failure_time_to_sorted_list(failure_time)
    end for
    i = 1
    j = 1
    while current_interval <= Tmax do
      if Parameter_i_fails and Consider_maintenance_activity_j_impact()
then
        if j <= M then
          Update_failure_time_of_parameter_i()
          j = j + 1
        end if
        Increment_failure_count_in_interval()
      else
        i = i + 1
      end if
      if i > Total_number_of_parameters then
        i = 1
      end if
      Move_to_next_interval()
    end while
  end while
  if Reached_set_number_of_simulations then
    Calculate_reliability_indices =
Compute_based_on_failure_counts_and_total_simulations()
  end if
end
```

## IV. ANALYZE

### A. Model analysis

In the process of fault diagnosis and reasoning, when confronted with the complexity of a virtual power plant system and assuming a failure within this system, Bayesian networks emerge as a powerful analytical tool. In this network, each node represents a possible event or state, while the connections between nodes reveal their causal relationships or correlations.

Specifically, when focusing on node F, which represents a failure in the virtual power plant system, the bucket elimination strategy can be employed to delve into the specific causes of the failure. This method, through logical reasoning, gradually eliminates impossible scenarios, thereby narrowing down the scope of the fault and enhancing the accuracy of diagnosis. By applying this approach and referring to the data provided in Table 3, it becomes evident that the conditional failure probability of the generator is significantly higher than that of other components, reaching approximately 0.627. This data objectively underscores the vulnerability of the generator within the virtual power plant system, indicating that it is the component with the highest probability of failure occurrence.

Based on this finding, a practical inference can be drawn: In actual operations, when a failure occurs in the virtual power plant system, to quickly locate and resolve the issue, the generator section should be inspected first, followed by other components such as circuit breakers and transformers.

TABLE III.     THE FAILURE PROBABILITY OF EACH COMPONENT IN THE CASE OF VIRTUAL POWER PLANT FAILURE

| Component | Circuit Breaker | Generator | Transformer | IED | Switch | Communication Line |
|---|---|---|---|---|---|---|
| Failure Probability | 0.362 | 0.627 | 0.127 | 0.010 | 0.003 | 0.016 |

### B. Example analysis

Assuming that the maximum operating time of the virtual power plant system is set at Tmax = 2000 hours, and this period is divided into multiple intervals, each with an interval of $\Delta T =$ 200 hours. To evaluate the reliability of this system, 1000 sampling simulation experiments were conducted. Each simulation recorded the failure time of the virtual power plant as well as the number of failures within each interval. Based on these data, reliability indicators of the virtual power plant were calculated and derived.

Figure 3 represents the failure probabilities of various components when the virtual power plant experiences a failure. The failure probability distribution depicted in Figure 3 shows a notable trend: During the first 200-hour interval after system startup, the failure probability exhibits an upward trend; however, as the operating time increases further, the failure probability gradually declines, indicating that the reliability of the virtual power plant is progressively improving.

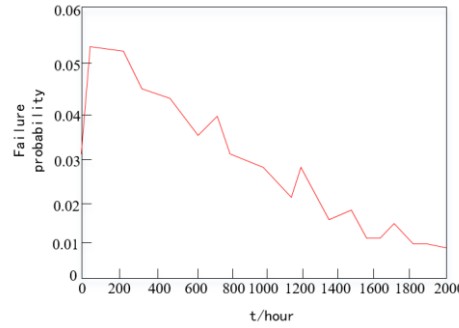

Fig 5.Virtual power plant failure probability distribution diagram

## V. CONCLUSION

This paper investigates six key factors that impact the reliability of virtual power plants. Firstly, a model based on Bayesian Networks is established to evaluate the reliability of virtual power plants. Subsequently, utilizing Monte Carlo simulation techniques, a quantitative analysis of the virtual power plant's reliability is conducted based on this Bayesian Network model.

(1) The failure probability distribution chart of the virtual power plant exhibits a notable trend: During the first 200 hours after system startup, the probability of failures increases; however, as the system's operating time extends, the failure probability gradually decreases, indicating a clear improvement in the reliability of the virtual power plant.

(2) When assessing the virtual power plant within the framework of Bayesian Networks, if the top event actually occurs, further analysis reveals that the generator section emerges as the most prone to failures, with a failure probability as high as 0.627. This discovery provides crucial insights for developing targeted fault prevention and maintenance strategies.

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
