# OpenReview forum: "Reliability Evaluation of Virtual Power Plants Based on Bayesian Networks"
_IEEE.org/ICIST/2024/Conference — IEEE ICIST 2024 Conference Submission_

### Official Review · Reviewer_87bc · 2024-08-21
**This paper investigates six key factors that impact the reliability of virtual power plants. This paper presents an interesting approach. However, the following comments should be considered in the revision.**

**Rating:** 7
**Confidence:** 3

**Review:**

Question 1:
Please provide more specific details regarding the shortcomings of traditional reliability analysis methods in evaluating virtual power plants (VPPs)? For instance, are there specific examples or case studies that illustrate these limitations? Additionally, the authors should clearly outline how the proposed Bayesian network approach addresses these shortcomings, especially in terms of intuitive representation of probabilistic relationships and integration of uncertain factors?
Question 2:
How effectively does the paper justify the selection and application of the Bayesian network approach for constructing the reliability model of VPPs? Are there discussions on alternative methods considered and reasons for their exclusion?
Question 3:
The paper should adequately highlight the novelty and significance of its contributions. Specifically, how does the proposed analysis model based on Bayesian networks advance the current understanding or methods used in assessing VPP reliability? Are there clear distinctions between this approach and previous studies, particularly in terms of the visualization of component influence and the combination with Monte Carlo simulation for comprehensive reliability analysis?

---

### Official Review · Reviewer_VD51 · 2024-08-25
**Accept, some mistakes should be revised**

**Rating:** 7
**Confidence:** 3

**Review:**

1. The font size of the formula is wrong which should be revised.
2. The contribution descriptions of this paper are missing and should be added to the Introduction.
3. The forms of Table II and Table III are unsightly, and should be modified.
4. The titles of Section II and Section IV are inaccurate and improper. Please revise them.

---

### Official Review · Reviewer_CdN5 · 2024-09-03
**This paper can be considered for publication.**

**Rating:** 6
**Confidence:** 2

**Review:**

The authors in this paper introduces the reliability evaluation of virtual power plants based on Bayesian networks. The reviewer has the following comments for this paper.
1. A contribution list should be added in the introduction part.
2. Some future investigation topics should be mentioned in the Conclusion part.

---

### Decision · Program_Chairs · 2024-09-06

Accept (Oral)